# Structure and Spectral Luminescence Properties of (ZrO_2_)_0.909_(Y_2_O_3_)_0.09_(Eu_2_O_3_)_0.001_ Ceramics Synthesized by Uniaxial Compaction and Slip Casting

**DOI:** 10.3390/ma15217722

**Published:** 2022-11-02

**Authors:** Mikhail Borik, Alexey Kulebyakin, Vladimir Kyashkin, Nataliya Larina, Elena Lomonova, Filipp Milovich, Valentina Myzina, Alexey Nezhdanov, Polina Ryabochkina, Nataliya Tabachkova, Efim Chernov

**Affiliations:** 1Prokhorov General Physics Institute of Russian Academy of Sciences, Vavilova Street, 38, 119991 Moscow, Russia; 2Institute of High Technologies and New Materials, National Research Ogarev Mordovia State University, Bolshevistskaya Street, 68, 430005 Saransk, Russia; 3Department of Materials Science, Moscow Polytechnic University, Bolshaya Semyonovskaya Street, 38, 107023 Moscow, Russia; 4Department of Materials Science of Semiconductors and Dielectrics, National University of Science and Technology «MISIS», Leninskiy Prospect, 4, 119049 Moscow, Russia; 5Physics Department, National Research Lobachevsky State University of Nizhny Novgorod, Gagarin Prospect, 23, 603022 Nizhny Novgorod, Russia; 6Research and Production Enterprise JSC «ECON», Lesnaya Street, 9, 249037 Obninsk, Russia

**Keywords:** zirconia, ceramic, single crystal, structure, luminescence spectra

## Abstract

The structure, phase composition and spectral luminescence properties of single crystal and ceramic specimens of (ZrO_2_)_0.909_(Y_2_O_3_)_0.09_(Eu_2_O_3_)_0.001_ solid solutions synthesized using uniaxial compaction and slip casting techniques have been compared. The ceramic specimens have been synthesized from crushed single crystal specimens of similar composition. It has been shown that the crystalline structures of the ceramic and single crystal specimens are identical and cubic. The ceramic specimens synthesized using different methods prove to have close microstructure patterns. The spectral luminescence properties of Eu^3+^ ions in the (ZrO_2_)_0.909_(Y_2_O_3_)_0.09_(Eu_2_O_3_)_0.001_ ceramic specimens are similar to those of the single crystals with similar composition. The (ZrO_2_)_0.909_(Y_2_O_3_)_0.09_(Eu_2_O_3_)_0.001_ ceramic specimens prove to have uncontrolled Cr^3+^:Al_2_O_3_ impurities due to the synthesis conditions.

## 1. Introduction

Zirconia-based materials exhibit a wide range of physicochemical properties that provide the possibility of their use in various applications [1,2,3]. For example, the high oxygen-ionic conductivity of these materials makes them suitable as solid-state electrolytes in solid oxide fuel cells, oxygen gages, oxygen pumps, etc. [3,4,5].

Zirconia-based solid state electrolytes are most typically synthesized using various ceramic technologies [6,7,8,9,10,11]. The synthesis method and conditions greatly affect the structural, physical, mechanical and transport properties of the ceramics. The structure, phase composition and density of the ceramic electrolytes depend largely on the precursor particle synthesis method. Ceramic material parameters such as grain size, porosity and component distribution homogeneity in the bulk and at the grain boundaries may dramatically change the electrophysical parameters of solid state electrolyte materials.

Precursor particles for ZrO_2_-Y_2_O_3_ ceramics are produced using solid state synthesis, laser ablation, hydrothermal techniques, sol–gel technologies, combustion, co-deposition of metal hydroxides, co-crystallization of salts, etc. The most widely used methods are solid state synthesis [12] and various modifications of sol–gel synthesis [13,14,15].

It was of interest within the scope of this work to implement zirconia ceramic synthesis from molten powder obtained by crushing of ZrO_2_-Y_2_O_3_ single crystals. The (ZrO_2_)_0.909_(Y_2_O_3_)_0.09_(Eu_2_O_3_)_0.001_ composition of the source single crystals was chosen because our earlier studies of the transport properties of ZrO_2_-Y_2_O_3_ crystals showed that crystals containing 9 mol.% Y_2_O_3_ stabilizing oxide have the highest ionic conductivity [16]. Eu^3+^ ions were introduced in the ZrO_2_-Y_2_O_3_ single crystals as a spectroscopic probe for studying the local neighborhood of the stabilizing oxide ions [17,18,19,20,21,22].

Thus, the aim of this work was to synthesize (ZrO_2_)_0.909_(Y_2_O_3_)_0.09_(Eu_2_O_3_)_0.001_ ceramics from crushed single crystals of similar composition by uniaxial compaction and slip casting, to study its structure, phase composition and the local neighborhood of the Eu^3+^ and Y^3+^ ions in this ceramic and to compare these parameters with those of single crystal specimens with similar composition.

## 2. Materials and Methods

The test (ZrO_2_)_0.909_(Y_2_O_3_)_0.09_(Eu_2_O_3_)_0.001_ solid solution single crystals were grown using directional melt crystallization at a 10 mm/h rate in a 130 mm diameter water-cooled crucible with direct high-frequency heating on a Kristall-407 plant (frequency 5.28 MHz, power 60 kW) (Moscow, Russia) [23,24]. The charge was prepared from zirconium oxide (ZrO_2_), yttrium oxide (Y_2_O_3_) and europium oxide (Eu_2_O_3_) with a purity of min. 99.96 wt.%. 

Some of the as-synthesized (ZrO_2_)_0.909_(Y_2_O_3_)_0.09_(Eu_2_O_3_)_0.001_ solid solution single crystals were mechanically crushed to powder less than 1 mm in size, which was used for the synthesis of ceramic specimens. The preliminarily crushed (ZrO_2_)_0.909_(Y_2_O_3_)_0.09_(Eu_2_O_3_)_0.001_ powder was ground with oleic acid addition in a drum with lining and grinding bodies made from stabilized zirconium dioxide (Y_2_O_3_ stabilizer) («Fritsch», Idar-Oberstein, Germany). The final powder had a grain size of ~40 µm and a specific surface area of ~8000 cm^2^/g.

The ceramic specimens were synthesized using two methods, i.e., uniaxial compaction and slip casting of thin films on a moving substrate.

Casting powders were prepared for the uniaxial compaction of ceramic specimens. Granulated powder was compacted in the form of discs (radius ~15 mm, thickness ~0.48 mm) at a 15 kN pressure. The specimens were air heat treated at 1680 °C for 2 h in furnaces with lanthanum chromite heaters and VK-97 closed corundum ceramic crucibles («Nabertherm», Lilienthal, Germany). The density of the specimens was ~5.85 g/cm^2^, i.e., 98% of the density of single crystals with similar composition (~5.98 g/cm^2^). The ceramic specimens synthesized by uniaxial compaction are shown in Figure 1a.

Slip casting of films on a moving ribbon (substrate) was implemented using a special plant designed by JSC «ECON» (JSC «ECON», Obninsk, Russia). The 7 mm × 7 mm × 0.3 mm pieces were placed on porous aluminum oxide kiln furniture and air heat treated at 1680 °C for 2 h. The density of the specimens was ~5.86 g/cm^2^, i.e., 98% of the density of single crystals with similar composition (~5.98 g/cm^2^). The ceramic specimens synthesized by slip casting are shown in Figure 1b.

The density of the (ZrO_2_)_0.909_(Y_2_O_3_)_0.09_(Eu_2_O_3_)_0.001_ single crystal and ceramic specimens was measured by hydrostatic weighing on a Sartogosm CE224-C balance (St. Petersburg, Russia).

The surface morphology and elemental composition of the ceramic specimens were studied using scanning electron microscopy (SEM) and energy dispersion spectroscopy under a Quanta TM 3D 200i scanning electron microscope equipped with a microanalysis system (EDS) (FEI Company, Hillsboro, OR, USA). The SEM images were taken at a 20 kV accelerating voltage under high vacuum conditions (~10^−3^ Pa).

The phase composition of the ceramic specimens was studied using X-ray diffraction on an Empyrean diffractometer from PANalytical B.V. Co. (CuKα radiation, λ = 1.5414 Å) with a vertical goniometer and a PIXcel 3D detector (PANalytical B.V. Co, Almelo, Netherlands). The diffraction patterns were identified using the JSPDS PDF 2 1911 database.

The phase composition was also studied using Raman spectroscopy on a NTEGRA SPECTRA instrument from NT-MDT Co. with a 632.8 nm He-Ne laser as the excitation source (NT-MDT Co., Zelenograd, Russia). The Raman spectra were recorded in the reflection scheme with a resolution of 0.8 cm^−1^.

The local neighborhood of the stabilizing oxide ions was studied using optical spectroscopy with Eu^3+^ ions being used as a spectroscopic probe. The T = 300 K luminescence spectra were recorded using an FHR 1000 spectrophotometer from Horiba Co. (Horiba Co., Kyoto, Japan) and a Hamamatsu R928B photomultiplier («Hamamatsu Photonics», Naka Ward, Japan) was used as a radiation detector. The excitation sources were YVO_4_:Nd (λ_exc._ = 532 nm) and LiYF_4_:Nd (λ_exc._ = 527 nm) lasers.

The 473 nm luminescence spectra were recorded for different areas of the ceramic specimen on a NTEGRA SPECTRA instrument from NT-MDT Co. (NT-MDT Co., Zelenograd, Russia). at room temperature. Confocal microscopic images were taken from 50 × 50 µm areas of the specimens.

The excitation spectra were recorded on a RF-5301PC spectrofluorometer from Shimadzu Co. (Shimadzu Co., Kyoto, Japan) with a 150 W xenon lamp used as an excitation source and a R212-14 photomultiplier used as a radiation detector.

## 3. Results

X-ray phase analysis data for the test (ZrO_2_)_0.909_(Y_2_O_3_)_0.09_(Eu_2_O_3_)_0.001_ ceramic specimens (Figure 2) suggest that the material is single-phase and has a fluorite-type cubic structure.

The test ceramic specimens synthesized using different methods (uniaxial compaction and slip casting) from crushed single crystals have close unit cell parameters, as summarized in Table 1. The lattice parameter of the (ZrO_2_)_0.909_(Y_2_O_3_)_0.09_(Eu_2_O_3_)_0.001_ ceramics is somewhat smaller than that of the single crystal material with similar composition.

The Raman spectra of the (ZrO_2_)_0.909_(Y_2_O_3_)_0.09_(Eu_2_O_3_)_0.001_ ceramics synthesized using different methods are shown in Figure 3. For comparison, the Raman spectrum of single crystals with similar composition is also shown.

The shape and positions of the bands in the Raman spectra of the ceramic specimens are close to those in the Raman spectra of the single crystals with similar composition. The Raman spectra of both the single crystal and ceramic specimens contain, along with the band typical of the cubic phase (c), a band near ~483 cm^−1^ that is typical of the t″ phase [25,26,27]. The structure of the t″ phase is close to cubic except for a small shift of the oxygen ions along a certain direction relative to their positions in the fluorite structure.

Figure 4 shows the surface SEM images of the test (ZrO_2_)_0.909_(Y_2_O_3_)_0.09_(Eu_2_O_3_)_0.001_ ceramic specimens.

The SEM data suggest that the microstructures of the ceramic specimens synthesized using different methods slightly differ. The ceramic specimens synthesized using uniaxial compaction (Figure 4a) have a rough surface. The grain sizes could only be determined after specimen polishing. On the contrary, the ceramic specimens synthesized using slip casting on a ribbon had a smoother surface and their microstructure could be visualized under optical or electron microscope without preliminary polishing. The grain size was approx. 10 to 40 μm for the specimens synthesized using different methods.

Energy dispersion elemental analysis of the ceramic materials suggests that their chemical composition is identical to that of the single crystal specimens except for the presence of aluminum on the specimen surfaces.

The presence and distribution of the uncontrolled aluminum oxide impurity was studied for different ceramic specimen areas. The content of europium oxide in the material is commensurable with the energy dispersion method error and therefore was not measured. Figure 5 shows the grain structure of the ceramic specimen synthesized using slip casting. Table 2 shows experimentally measured concentrations of ZrO_2_, Y_2_O_3_ and Al_2_O_3_ oxides in this ceramic specimen for the areas shown in Figure 5.

Table 2 suggests that the Al_2_O_3_ concentration at the ceramic grain surfaces was 0.6 mol.%. The oxide distribution along the grain boundaries was local (the Al_2_O_3_ concentration being within 1–3 mol.%). In the area corresponding to the intergrain volume (area 1 in Figure 5) between three ceramic grains, the Al_2_O_3_ concentration was 3.3 mol.%.

It can be hypothesized that the origin of the aluminum impurity is the corundum crucible that the specimens contacted during annealing. Analysis of literary data reveals evidence of the low solubility of aluminum oxide in ZrO_2_-Y_2_O_3_. It was reported [28,29,30,31] that the solubility limit of Al_2_O_3_ in ZrO_2_-9 mol.% Y_2_O_3_ at T = 1700 °C is 0.7 mol.%. At higher Al_2_O_3_ concentrations, aluminum oxide in the ZrO_2_-Y_2_O_3_ ceramics is present in the form of particles located inside the grains and at the grain boundaries or forms grain-boundary phases with high Al_2_O_3_ content.

Taking into account the results of this work and earlier studies [28,29,30,31], one can conclude that some of the Al_2_O_3_ impurity is dissolved in the ceramics whereas most of the impurity is localized at the grain boundaries and, to a smaller extent, on the ceramic surface in the form of inclusions. Al_2_O_3_ solubility in (ZrO_2_)_0.909_(Y_2_O_3_)_0.09_(Eu_2_O_3_)_0.001_ ceramic solid solutions can be the cause of the decrease in the lattice parameter of the solid solutions in comparison to the single crystals of similar composition.

Figure 6 shows the luminescence spectra of the (ZrO_2_)_0.909_(Y_2_O_3_)_0.09_(Eu_2_O_3_)_0.001_ single crystal and ceramic solid solutions recorded with excitation of the ^5^D_1_ level of the Eu^3+^ ions by a 532 nm laser at room temperature.

Analysis of the luminescence spectra for the transitions between the ^5^D_0_ and ^7^F_J_(_0-4_) multiplets of the Eu^3+^ ions in the (ZrO_2_)_0.909_(Y_2_O_3_)_0.09_(Eu_2_O_3_)_0.001_ ceramic solid solutions recorded with excitation of the ^5^D_1_ level of the Eu^3+^ ions by a 532 nm laser at T = 300 K shows that they contain the same spectral bands as the respective spectra of the (ZrO_2_)_0.909_(Y_2_O_3_)_0.09_(Eu_2_O_3_)_0.001_ single crystal solid solutions. The luminescence spectra of the (ZrO_2_)_0.909_(Y_2_O_3_)_0.09_(Eu_2_O_3_)_0.001_ single crystals for the ^5^D_0_→^7^F_0-2_ transitions of the Eu^3+^ ions, which are superimpositions of the bands of the Eu^3+^ optical centers with different neighborhoods, were reported earlier [17]. The numbers I, II and IV’ in Figure 6 mark the respective Eu^3+^ optical centers. Optical center I is where the Eu^3+^ ions have one oxygen vacancy and are surrounded by seven oxygen atoms. Optical center II is where the Eu^3+^ ions do not have oxygen vacancies in the first coordination shell but have one oxygen vacancy in the second coordination shell. Optical center IV’ represents the centers in which oxygen vacancies are located in the farthest coordination shells of the Eu^3+^ ions.

It can be seen from Figure 6 that the intensity ratio of the spectral bands for the ^5^D_0_→^7^F_0-2_ transitions of the Eu^3+^ ions pertaining to different optical centers in the synthesized (ZrO_2_)_0.909_(Y_2_O_3_)_0.09_(Eu_2_O_3_)_0.001_ ceramic specimens is the same as the respective intensity ratio of the optical centers for the single crystal material of similar composition. The difference in the luminescence spectra for the ceramics is the presence of two intense and narrow bands near 693 and 694.5 nm and a change in the shape of the spectral bands in the 665–730 nm region. Taking into account the results of earlier studies [17,18,19,20,21,22], one can conclude that these changes in the luminescence spectra of the ceramics are not caused by the optical transitions of the Eu^3+^ ions. It can also be seen from Figure 6 that the bands near 693 and 694.5 nm for different ceramic specimens have different intensities relative to the luminescence bands of the Eu^3+^ ions.

Figure 7 shows the luminescence spectra of the (ZrO_2_)_0.909_(Y_2_O_3_)_0.09_(Eu_2_O_3_)_0.001_ single crystal and ceramic solid solutions recorded with excitation of the ^5^D_1_ level of the Eu^3+^ ions by a 527 nm laser at room temperature.

Analysis of the luminescence spectra of the Eu^3+^ ions in the (ZrO_2_)_0.909_(Y_2_O_3_)_0.09_(Eu_2_O_3_)_0.001_ single crystal and ceramic solid solutions shown in Figure 7 shows that the luminescence spectra of the ceramic specimens are identical to each other and to the luminescence spectrum of the single crystal specimen. The luminescence spectra of the ceramic specimens differ from those of the single crystal specimen through the presence of two narrow bands at 693 and 694.5 nm whose intensities differ for the two ceramic specimens. It should be noted that the intensities of the bands in the luminescence spectrum of the Eu^3+^ ions under excitation by a 527 nm laser are lower than the intensities of the bands in the luminescence spectrum recorded with 532 nm excitation.

Analysis of literary data shows that the Cr^3+^: Al_2_O_3_ luminescence spectrum excited by a 532 nm laser contains two intense and narrow bands near 692.8 and 694.2 nm (the so-called R-bands) caused by the ^2^E→^4^A_2_ transition of the Cr^3+^ ions and also contains several additional weak bands near 660, 670, 706 and 714 nm (the N-bands and the sidebands) [32,33,34]. The positions of these bands are similar to those of the bands found in the luminescence spectra of the (ZrO_2_)_0.909_(Y_2_O_3_)_0.09_(Eu_2_O_3_)_0.001_ ceramics (Figure 6 and Figure 7). Thus, all the ceramic specimens contained a chromium impurity along with aluminum oxide, as suggested by the presence of additional bands in the luminescence spectra of the ceramic specimens.

The chromium impurity probably originates from the use of lanthanum chromite (LaCrO_3_) heaters during specimen heat treatment in insufficiently closed aluminum oxide (Al_2_O_3_) crucibles. Since the ceramic specimens were sintered at a high temperature (1680 °C), chromium ions could evaporate from the heater material LaCrO_3_. The volatility of chromium in LaCrO_3_ at high temperatures is a well-known problem [35]. It was hypothesized that chromium evaporation occurs by the following reaction:(1)$$2{\mathrm{LaCrO}}_{3}\left(s\right)={\mathrm{La}}_{2}O_{3}\left(s\right)+{\mathrm{Cr}}_{2}O_{3}\left(g\right)$$

Chromium oxide may also reduce to metallic chromium at above 1200 °C [36].

According to literary data, Cr_2_O_3_ solubility in Al_2_O_3_ is unlimited [37], and given that the luminescence spectra (Figure 6 and Figure 7) only contain bands typical of the Cr^3+^ ions in Al_2_O_3_, one can conclude that Cr_2_O_3_ interacts mainly with Al_2_O_3_ to form the Cr_2_O_3_-Al_2_O_3_ solid solution. The chromium impurity seems to not enter into the ZrO_2_-Y_2_O_3_-Eu_2_O_3_ solid solution since the experimental luminescence spectra did not contain the wide band in the 700–1100 nm region which is typical of Cr^3+^ ion luminescence in ZrO_2_-Y_2_O_3_ [38,39]. Furthermore, Cr_2_O_3_ solubility in ZrO_2_-Y_2_O_3_ was quite low, only 0.7 mol.% for the ZrO_2_-8 mol.% Y_2_O_3_ ceramics at T = 1450 °C [36,40].

An inhomogeneous distribution of the uncontrolled impurity in the ceramic specimens is suggested by the change in the R_1_ and R_2_ luminescence band intensities of the Cr^3+^ ions in Al_2_O_3_ in comparison to the luminescence bands of the Eu^3+^ ions in different ceramic surface areas (Figure 8 and Figure 9). The R_1_ and R_2_ luminescence band intensity distribution map for the Cr^3+^ ions in Al_2_O_3_ in a 50 μm × 50 μm surface area (Figure 8a and Figure 9a) is shown in Figure 8b and Figure 9b. Figure 8c and Figure 9c show the luminescence spectra for different surface areas of the test ceramics excited with a 473 nm laser. Figure 8d and Figure 9d separately show the R_1_ and R_2_ luminescence bands of the Cr^3+^ ions in Al_2_O_3_ for the ^2^E→^4^A_2_ transition.

These spectra are shown in relative units normalized to unity relative to the luminescence band peaking at 606.4 nm caused by the ^5^D_0_→^7^F_2_ transition of the Eu^3+^ ions. The color of the R_1_ and R_2_ luminescence bands of the Cr^3+^ ions in Figure 8c,d and Figure 9c,d corresponds to the color in the scale shown to the right of the intensity distribution maps in Figure 8b and Figure 9b characterizing the intensity of these bands.

Thus, analysis of the R_1_ and R_2_ luminescence band intensity distribution maps for the Cr^3+^ ions in Al_2_O_3_ on the ceramic specimen surfaces, as shown in Figure 8b and Figure 9b, showed that the uncontrolled Cr^3+^: Al_2_O_3_ impurities in the test ceramic specimens were in the form of discrete inclusions.

Figure 10 shows the luminescence excitation spectra of (ZrO_2_)_0.879_(Y_2_O_3_)_0.12_(Eu_2_O_3_)_0.001_ single crystal and (ZrO_2_)_0.909_(Y_2_O_3_)_0.09_(Eu_2_O_3_)_0.001_ ceramic solid solutions. The recording wavelength of 606 nm corresponds to the position of the most intense ^5^D_0_→^7^F_2_ transition band of the Eu^3+^ ions.

The excitation spectra of the (ZrO_2_)_0.909_(Y_2_O_3_)_0.09_(Eu_2_O_3_)_0.001_ ceramics (Figure 10) contain a sufficiently wide symmetrical band in the 220 to 290 nm region peaking at ~253 nm, which according to earlier data [41] originates from the O^2−^→Eu^3+^ charge transfer transition. For the (ZrO_2_)_0.879_(Y_2_O_3_)_0.12_(Eu_2_O_3_)_0.001_ single crystal, this band is at a greater wavelength (~265 nm). Along with the O^2−^→Eu^3+^ charge transfer band, the luminescence excitation spectra of the (ZrO_2_)_0.879_(Y_2_O_3_)_0.12_(Eu_2_O_3_)_0.001_ single crystal and (ZrO_2_)_0.909_(Y_2_O_3_)_0.09_(Eu_2_O_3_)_0.001_ ceramic solid solutions contain bands corresponding to intraconfigurational f-f transitions between the multiplets of the ^7^F main state and the ^5^D, ^5^L and ^5^G excited states of the Eu^3+^ ions. The most intense band is the one at 528 nm that pertains to the ^7^F_0, 1_→^5^D_2_ transition. The difference in the spectral band intensities between the single crystals and the ceramics for the group of optical transitions in the 380–430 nm region seems to be caused by stronger scattering from the non-transparent ceramic specimens.

## 4. Discussion

In this work ceramic specimens were synthesized using uniaxial compaction and slip casting from powders of crushed (ZrO_2_)_0.909_(Y_2_O_3_)_0.09_(Eu_2_O_3_)_0.001_ single crystals grown by directional melt crystallization in a cold skull. The ceramic specimens were sintered in air at 1680 °C for 2 h. The density of the as-synthesized ceramic specimens was 98% of the density of the single crystal specimens.

The (ZrO_2_)_0.909_(Y_2_O_3_)_0.09_(Eu_2_O_3_)_0.001_ ceramic solid solutions synthesized using different methods from crushed single crystals are single-phase and have a fluorite-type cubic structure with close lattice parameters. For uniaxially compacted ceramics, a = 5.1381(3) Å; for slip cast ceramics, a = 5.1368(3) Å. Raman spectroscopy showed that the structure of the test ceramic specimens consists of a t″ phase.

The surface morphology of the ceramic specimens synthesized using different methods differs, but the microstructures of the specimens are quite similar. The (ZrO_2_)_0.909_(Y_2_O_3_)_0.09_(Eu_2_O_3_)_0.001_ slip cast ceramic specimens have more homogeneous surfaces. The grain size of the ceramic specimens was approx. 10 to 40 µm.

Comparison of the luminescence spectra of the Eu^3+^ ions in the (ZrO_2_)_0.909_(Y_2_O_3_)_0.09_(Eu_2_O_3_)_0.001_ ceramic and single crystal solid solutions did not reveal any tangible difference. The luminescence excitation spectra of the Eu^3+^ ions in the (ZrO_2_)_0.909_(Y_2_O_3_)_0.09_(Eu_2_O_3_)_0.001_ ceramics are similar in band shape and position to the luminescence excitation spectra of the Eu^3+^ ions in the single crystals except for the O^2−^→ Eu^3+^ charge transfer band’s shift towards smaller wavelengths.

Elemental analysis revealed the presence of aluminum impurities, and optical spectroscopic study of the (ZrO_2_)_0.909_(Y_2_O_3_)_0.09_(Eu_2_O_3_)_0.001_ ceramic specimens showed the presence of uncontrolled Cr^3+^:Al_2_O_3_ impurities originating from conditions related to the synthesis process.

## Figures and Tables

**Figure 1 materials-15-07722-f001:**
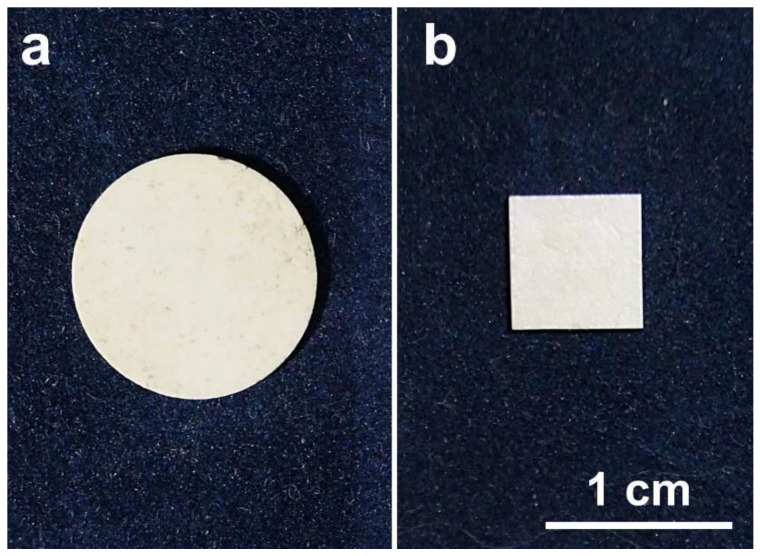
(ZrO_2_)_0.909_(Y_2_O_3_)_0.09_(Eu_2_O_3_)_0.001_ ceramic specimens synthesized from crushed single crystals of similar composition using (**a**) uniaxial compaction and (**b**) slip casting.

**Figure 2 materials-15-07722-f002:**
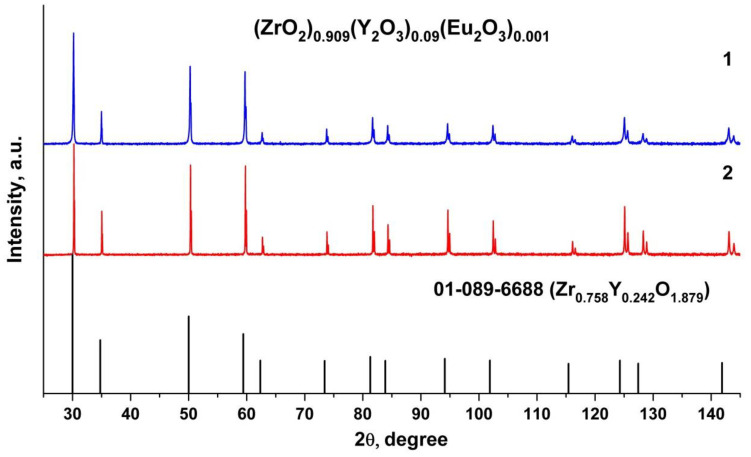
Fragments of the X-ray diffraction patterns of (ZrO_2_)_0.909_(Y_2_O_3_)_0.09_(Eu_2_O_3_)_0.001_ ceramic solid solutions synthesized using different methods from crushed single crystals: (1) uniaxial compaction; (2) slip casting.

**Figure 3 materials-15-07722-f003:**
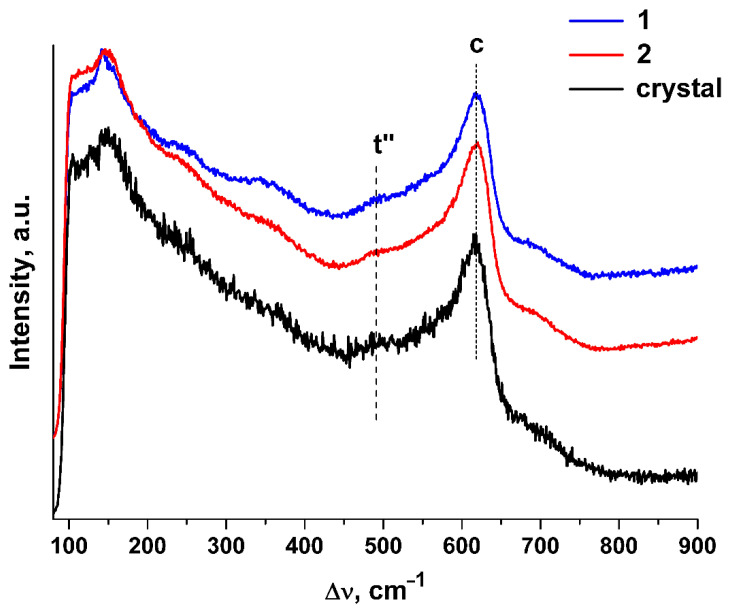
Raman spectra of (ZrO_2_)_0.909_(Y_2_O_3_)_0.09_(Eu_2_O_3_)_0.001_ single crystal and ceramic solid solutions synthesized using different methods from crushed single crystals: (1) uniaxial compaction; (2) slip casting. λ_exc._ = 632.8 nm, T = 300 K.

**Figure 4 materials-15-07722-f004:**
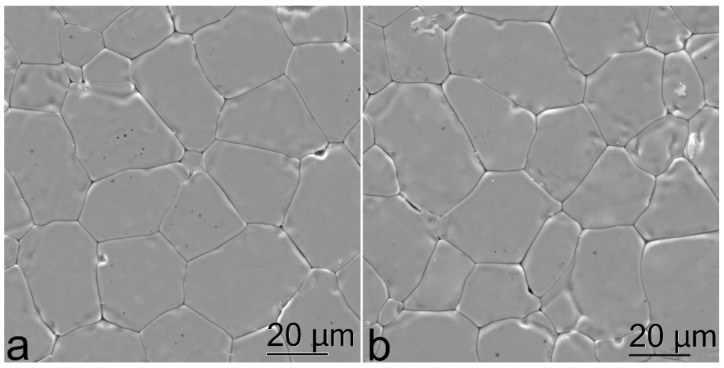
Microstructure images of the (ZrO_2_)_0.909_(Y_2_O_3_)_0.09_(Eu_2_O_3_)_0.001_ ceramic specimens synthesized using different methods from crushed single crystals: (**a**) uniaxial compaction; (**b**) slip casting.

**Figure 5 materials-15-07722-f005:**
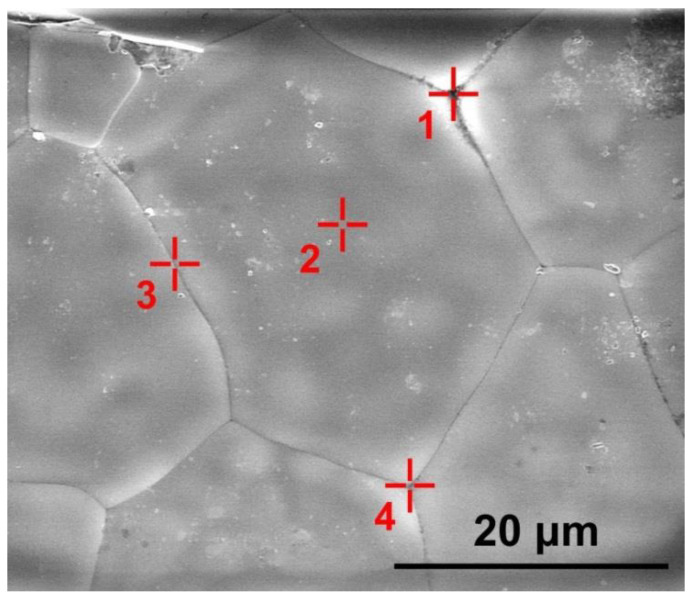
SEM image of the (ZrO_2_)_0.909_(Y_2_O_3_)_0.09_(Eu_2_O_3_)_0.001_ ceramic specimen synthesized using slip casting. The 
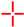
 symbols show areas (1, 2, 3, 4) in which the ZrO_2_, Y_2_O_3_ and Al_2_O_3_ oxide concentrations were measured.

**Figure 6 materials-15-07722-f006:**
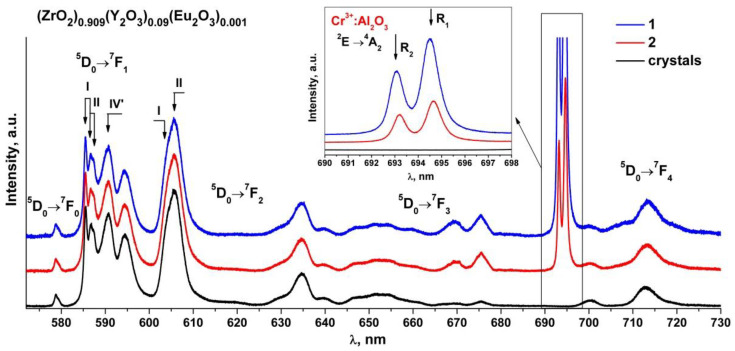
Luminescence spectra of (ZrO_2_)_0.909_(Y_2_O_3_)_0.09_(Eu_2_O_3_)_0.001_ single crystal and ceramic solid solutions synthesized using different methods from crushed single crystals: (1) uniaxial compaction; (2) slip casting. λ = 532 nm, T = 300 K. Inset: luminescence bands (R_1_ and R_2_) of Cr^3+^ ions in Al_2_O_3_ for the ^2^E→^4^A_2_ transition.

**Figure 7 materials-15-07722-f007:**
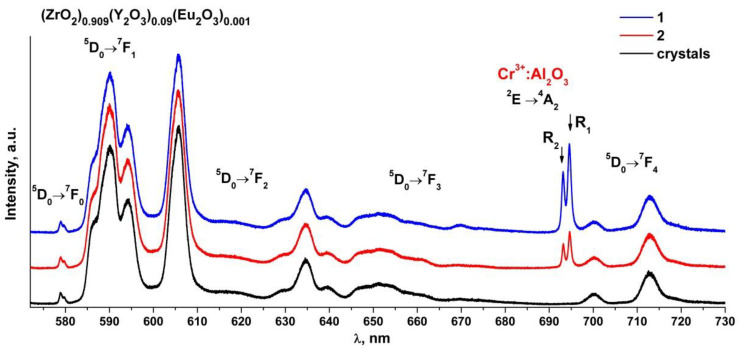
Luminescence spectra of (ZrO_2_)_0.909_(Y_2_O_3_)_0.09_(Eu_2_O_3_)_0.001_ single crystal and ceramic solid solutions synthesized using different methods from crushed single crystals: (1) uniaxial compaction; (2) slip casting. λ = 527 nm, T = 300 K.

**Figure 8 materials-15-07722-f008:**
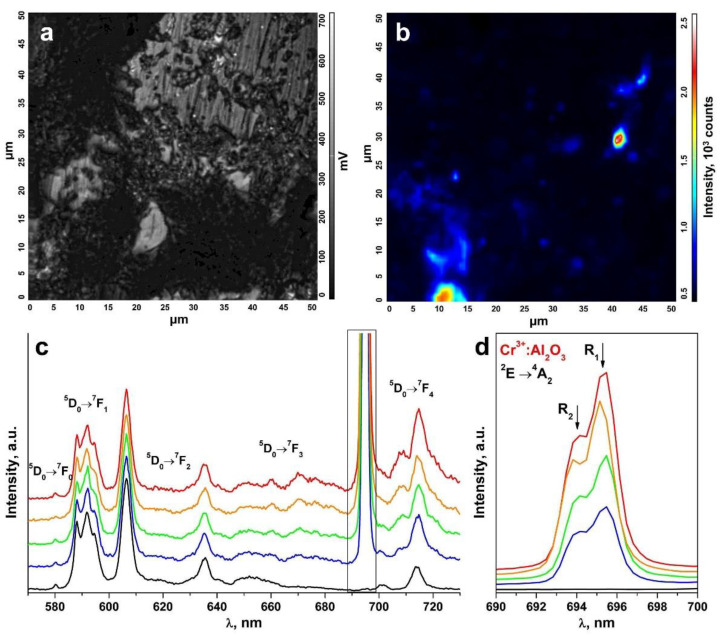
(**a**) Surface image of a ceramic specimen synthesized using uniaxial compaction; (**b**) R_1_ and R_2_ luminescence band intensity distribution map for the Cr^3+^ ions in Al_2_O_3_ on the surface area shown in Figure 8a; (**c**) luminescence spectra of the ceramic specimen excited with a λ = 473 nm laser (T = 300 K) corresponding to the intensity scale in Figure 8b; (**d**) R_1_ and R_2_ luminescence bands of the Cr^3+^ ions in Al_2_O_3_ for the ^2^E→^4^A_2_ transition.

**Figure 9 materials-15-07722-f009:**
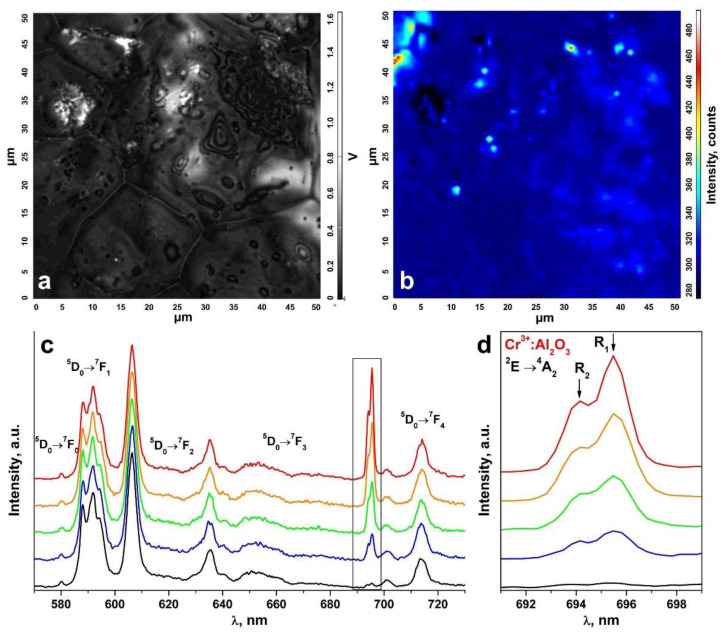
(**a**) Surface image of a ceramic specimen synthesized using slip casting; (**b**) R_1_ and R_2_ luminescence band intensity distribution map for the Cr^3+^ ions in Al_2_O_3_ on the surface area shown in Figure 9a; (**c**) luminescence spectra of the ceramic specimen excited with a λ = 473 nm laser (T = 300 K) corresponding to the intensity scale in Figure 9b; (**d**) R_1_ and R_2_ luminescence bands of the Cr^3+^ ions in Al_2_O_3_ for the ^2^E→^4^A_2_ transition.

**Figure 10 materials-15-07722-f010:**
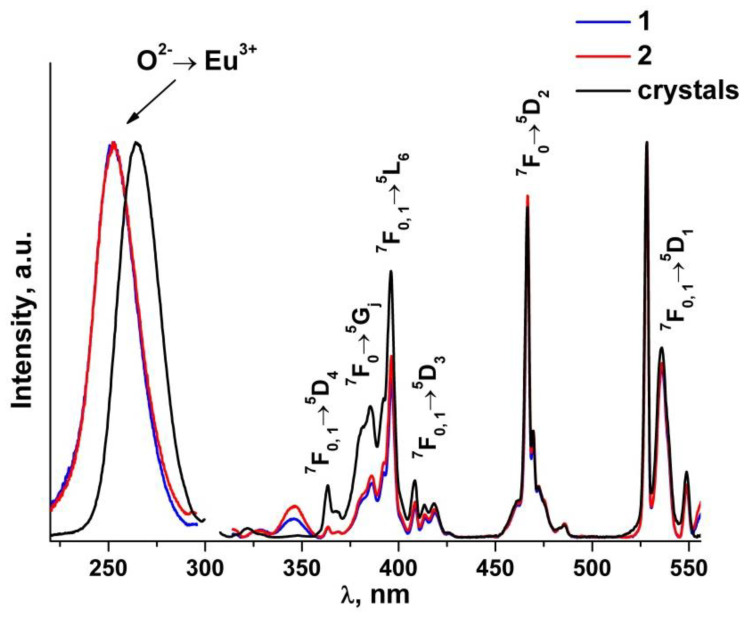
Luminescence excitation spectra of (ZrO_2_)_0.879_(Y_2_O_3_)_0.12_(Eu_2_O_3_)_0.001_ single crystal and (ZrO_2_)_0.909_(Y_2_O_3_)_0.09_(Eu_2_O_3_)_0.001_ ceramic solid solutions synthesized using different methods from crushed single crystals: (1) uniaxial compaction; (2) slip casting. λ_em_ = 606 nm, T = 300 K.

**Table 1 materials-15-07722-t001:** Phase composition and lattice parameters of (ZrO_2_)_0.909_(Y_2_O_3_)_0.09_(Eu_2_O_3_)_0.001_ single crystals and ceramics synthesized using different methods from crushed single crystals.

(ZrO_2_)_0.909_(Y_2_O_3_)_0.09_(Eu_2_O_3_)_0.001_ Solid Solution	Phase Composition ^1^	Space Group	Unit Cell Parameters
			a, Å
Single Crystal	c-ZrO_2_	Fm3m	5.141(1) [17]
Ceramic #1 (Uniaxial Compaction)	c-ZrO_2_	Fm3m	5.1381(3)
Ceramic #2 (Slip Casting)	c-ZrO_2_	Fm3m	5.1368(3)

^1^ c is cubic ZrO_2_ modification.

**Table 2 materials-15-07722-t002:** Elemental analysis data for (ZrO_2_)_0.909_(Y_2_O_3_)_0.09_(Eu_2_O_3_)_0.001_ ceramics synthesized using slip casting.

Area	Concentration, mol.%		
	ZrO_2_	Y_2_O_3_	Al_2_O_3_
1	87.1	9.6	3.3
2	89.6	9.9	0.6
3	89.2	10.0	0.8
4	89.2	9.9	0.9

## Data Availability

All the data are available within the manuscript.

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
