# Peer review of "Structure and Spectral Luminescence Properties of (ZrO2)0.909(Y2O3)0.09(Eu2O3)0.001 Ceramics Synthesized by Uniaxial Compaction and Slip Casting"

_materials, 2022, doi:10.3390/ma15217722_

Round 1

Reviewer 1 Report

The aim of present work was to study of Structure and Spectral Luminescence Properties of (ZrO2)0,909(Y2O3)0,09(Eu2O3)0,001 Ceramics Synthesized by Uniaxial Compaction and Slip Casting.

The manuscript is very well written and accurate. The results of the study are interesting.

Therefore, this manuscript is recommended for publication after addressing the following items:

1) X-ray diffraction patterns (Fig 2)  are sufficient to show up to 100 degree 2 Theta.

2) Were the XRD line shifts of the solid solutions relative to the matrix recorded ?

3) I recommend determining the sizes of crystallites of solid solutions by using Scherrer or Williamson-Hall method and comparing them with the sizes determined by the microscope.

4) The Uv-Vis-DRS research would be very interesting. I would suggest evaluate the energy gap for the obtained phases (Kubelka-Munk vs eV).

Author Response

We would like to thank the reviewers for their time and relevant comments.

Reviewer №1

The aim of present work was to study of Structure and Spectral Luminescence Properties of (ZrO2)0,909(Y2O3)0,09(Eu2O3)0,001 Ceramics Synthesized by Uniaxial Compaction and Slip Casting.

The manuscript is very well written and accurate. The results of the study are interesting.

Therefore, this manuscript is recommended for publication after addressing the following items:

1) X-ray diffraction patterns (Fig 2) иare sufficient to show up to 100 degree 2 Theta.

2) Were the XRD line shifts of the solid solutions relative to the matrix recorded?

3) I recommend determining the sizes of crystallites of solid solutions by using Scherrer or Williamson-Hall method and comparing them with the sizes determined by the microscope.

4) The Uv-Vis-DRS research would be very interesting. I would suggest evaluate the energy gap for the obtained phases (Kubelka-Munk vs eV).

Answers

1). The shift of the diffraction lines is better seen at large angles 2θ. Therefore, in Fig. 2, the diffraction patterns are presented in a wide range of 2θ.

2) As can be seen from Table 1, the lattice parameters of ceramic samples from crushed single crystals differ from the lattice parameter of a single crystal.

3) Following your recommendations, we calculated the sizes of the coherent scattering regions of ceramics using the Williamson-Hall method. They were about 100 nm, which is much smaller than the grain sizes determined from SEM images.

4) We have not done UV-Vis-DRS studies. We agree that this technique is of some interest and will take this into account when conducting further research.

Reviewer 2 Report

The article by Borik et al shows interesting results concerning physico-chemical and spectroscopic properties of ZrO2-Y2O3 ceramics doped with Eu3+ ions prepared from preliminary grown single crystals using two different techniques as uniaxial compaction and slip casting.

In my opinion this work deserves to be published since these original results are correctly presented and discussed.

I do not find weaknesses in this article. However some comments should be addressed to authors as follows:

- Fig. 3 raman spectra: the bands near 160 cm-1 are not commented.

- Line 244: if I understood correctly LaCrO3 is used as heater in the uniaxial compaction technique and not for the slip casting one. Am I wrong? Anyway Cr3+ ions can occur from alumina crucibles or alumina furniture for the slip casting method. Can you discuss briefly on the difference of Cr3+ emission intensity for both ceramics 1 and 2?

- Fig. 10: concerning the Eu-O CTS  a blue shift is observed for both ceramics in comparison with single crystals excitation spectrum. Can you discuss this result? Is it due to the lowering of particle size?

Author Response

We would like to thank the reviewers for their time and relevant comments.

Reviewer №2

The article by Borik et al shows interesting results concerning physico-chemical and spectroscopic properties of ZrO2-Y2O3 ceramics doped with Eu3+ ions prepared from preliminary grown single crystals using two different techniques as uniaxial compaction and slip casting.

In my opinion this work deserves to be published since these original results are correctly presented and discussed.

I do not find weaknesses in this article. However, some comments should be addressed to authors as follows:

1) Fig. 3 Raman spectra: the bands near 160 cm-1 are not commented.

2) Line 244: if I understood correctly LaCrO3 is used as heater in the uniaxial compaction technique and not for the slip casting one. Am I wrong? Anyway Cr3+ ions can occur from alumina crucibles or alumina furniture for the slip casting method. Can you discuss briefly on the difference of Cr3+ emission intensity for both ceramics 1 and 2?

3) Fig. 10: concerning the Eu-O CTS  a blue shift is observed for both ceramics in comparison with single crystals excitation spectrum. Can you discuss this result? Is it due to the lowering of particle size?

Answer

1) A low-intensity broad line near 160 cm-1 in the Raman spectra, according to the results of [Hemberger, Y.; Wichtner, N.; Berthold, C.; Nickel, K.G. Quantification of yttria in stabilized zirconia by Raman spectroscopy. Int. J. Appl. Ceram. Technol. 2016, 13, 116–124.], characterizes the mode of vibrations of the cubic and tetragonal phases in solid solutions based on zirconium dioxide.

2) Lanthanum chromite (LaCrO3) heaters were used in furnaces in which heat treatment of all ceramic samples made both by uniaxial pressing and slip casting was carried out. The intensity of the luminescence lines of Cr3+ ions for ceramics obtained by various methods can apparently be due to different concentrations of the uncontrolled Cr3+ impurity in the samples. However, due to the uneven distribution of Cr3+ on the surface and in the bulk of the samples, a quantitative comparison of the Cr3+ content from the luminescence spectra seems to be incorrect.

3) We assume that the difference in the positions of the maxima of the band associated with the O2- --> Eu3+ charge transfer in the excitation spectra is due to the peculiarities of the interaction of oxygen vacancies. However, an unequivocal answer to this question requires further research.

Round 2

Reviewer 1 Report

Accept in present form